# Post-supereruption recovery at Toba Caldera

Adonara E. Mucek[1], Martin Danišík[2], Shanaka L. de Silva[1], Axel K. Schmitt[3], Indyo Pratomo[4] & Matthew A. Coble[5]

Large calderas, or supervolcanoes, are sites of the most catastrophic and hazardous events on Earth, yet the temporal details of post-supereruption activity, or resurgence, remain largely unknown, limiting our ability to understand how supervolcanoes work and address their hazards. Toba Caldera, Indonesia, caused the greatest volcanic catastrophe of the last 100 kyr, climactically erupting ∼74 ka. Since the supereruption, Toba has been in a state of resurgence but its magmatic and uplift history has remained unclear. Here we reveal that new [14]C, zircon U–Th crystallization and (U–Th)/He ages show resurgence commenced at 69.7 ± 4.5 ka and continued until at least ∼2.7 ka, progressing westward across the caldera, as reflected by post-caldera effusive lava eruptions and uplifted lake sediment. The major stratovolcano north of Toba, Sinabung, shows strong geochemical kinship with Toba, and zircons from recent eruption products suggest Toba's climactic magma reservoir extends beneath Sinabung and is being tapped during eruptions.

[1] College of Earth, Ocean, and Atmospheric Sciences, Oregon State University, 104 CEOAS Administration Building, Corvallis, OR 97331-5503, USA. [2] John de Laeter Centre, Building 301, Curtin University, Bentley, Western Australia 6845, Australia. [3] Institute of Earth Sciences, Ruprecht-Karls-Universität Heidelberg, Im Neuenheimer Feld 236, Heidelberg D-69120, Germany. [4] Geological Agency of Indonesia, JL. Diponegoro No. 57, Jawa Barat, Indonesia. [5] SHRIMP-RG Lab, Green Earth Sciences Building, 367 Panama Street Room 89, Stanford University, California 94305, USA. Correspondence and requests for materials should be addressed to A.E.M. (email: muceka@geo.oregonstate.edu).

Many of Earth's largest calderas, or supervolcanoes, like Yellowstone (Wyoming, USA) and the Valles Caldera (New Mexico, USA), record multi-million year histories punctuated by geologically instantaneous and catastrophic super-eruptions[1–3]. During the intervening periods of relative quiescence, the caldera re-establishes magmastatic, isostatic and lithostatic equilibrium after the catastrophic disruption of the caldera-forming (climactic) eruption. Referred to as resurgence, this period of post-climactic recovery is recognized as one of the most dynamic phases of a caldera cycle during which uplift of the caldera floor is accompanied by further eruptions, establishment of hydrothermal and geothermal systems, and ore formation[4–6]. However, despite the recognition that all currently active large calderas are in resurgence with the potential for significant further hazard[7], precise timing of events during resurgence at large calderas are unknown, impeding our ability to fully assess any accompanying hazard. One of the main challenges is to unravel how activity varies in space and time, but to date, efforts have been hindered by the limitations of the available techniques[8–10]. Here we show how the application of combined $^{238}U$–$^{230}Th$ disequilibrium and (U–Th)/He dating of zircon crystals[11–14] from eruptions at Toba Caldera and $^{14}C$ ages of lake sediment deposits found on lava extrusions reveal for the first time details of the post-caldera eruption and uplift history that informs the potential hazards of the largest young supervolcano on Earth, which remain heavily debated[15–18]. We confirm that Toba Caldera formed during the climactic eruption ∼74 ka but find that the resurgence started within a few 1,000 years after caldera formation, and continued until at least ∼2.7 ka. Furthermore, Sinabung, a currently active volcano north of Toba, shows similar $^{87}Sr$/$^{86}Sr$ isotopic values to Toba's rocks, unlike any other arc volcano on Sumatra, and also has an overlapping zircon crystallization history based on $^{238}U$–$^{230}Th$ disequilibrium age constraints. This raises the possibility that Sinabung and Toba share a common magmatic source requiring reconsideration of the potential hazard to this region.

## Results

**Eruptions and resurgence at Toba.** Eruptive activity at Toba began ∼1.2 Ma with the production of the Haranggaol Dacite Tuff, followed by caldera-forming eruptions at ∼0.8 Ma (the Oldest Toba Tuff) and ∼0.5 Ma (the Middle Toba Tuff)[10]. Toba's third and last caldera-forming eruption, that of the climactic Youngest Toba Tuff (YTT), occurred ∼74 ka (ages range from 73.9 ± 0.6 (ref. 19) to 75.0 ± 1.8 ka (ref. 20)) and created the present lake basin by the collapse of the roof of the magma reservoir[10]. The lake formed within ∼1,500 years[10] and over the next 66 kyr at least, episodic trapdoor-style uplift of the caldera floor formed Samosir Island[21]. Several lava effusions accompanied resurgent uplift and faulting[22] on the margins of Samosir and the caldera (Fig. 1), but to date, $^{40}Ar$/$^{39}Ar$ age determinations have not been able to differentiate their ages from the age of the climactic eruption despite clear stratigraphic evidence of their relative youth[10].

**Zircon eruption ages.** In this work, eruption ages for ten samples from Toba Caldera have been determined using the combined $^{238}U$–$^{230}Th$ disequilibrium and (U–Th)/He zircon dating method (Table 1; Supplementary Data 1; Methods section). The eruption ages for the YTT (79.5 ± 5.5 and 78.1 ± 5.8 ka; Fig. 2a,b) determined by this approach overlap within 2 s.e. ($\sigma$) uncertainty with the recent $^{40}Ar$/$^{39}Ar$ ages of 73.9 ± 0.6 (ref. 19) and 75.0 ± 1.8 ka (ref. 20; Fig. 2k), indicating concordance of the results and providing a cross check for the (U–Th)/He eruption ages. In contrast, ages for the post-caldera lava domes along the eastern

coast of Samosir Island range from 69.7 ± 4.5 to 65.3 ± 4.6 ka (Fig. 2e–h). Younger eruption ages have been determined for centres along the western rim of Toba Caldera where the Pardepur lava domes, at 56.9 ± 3.9 and 63.4 ± 5.6 ka (Fig. 2i,j), and the Pusuk Buhit lava flows, at 62.2 ± 7.1 and 54.5 ± 8.0 ka (Fig. 2c,d), are located. These data suggest that post-YTT eruptive activity started immediately after the climactic eruption and continued episodically for at least 15.2 ± 9.2 kyr thereafter (Fig. 2k).

Age and stratigraphic relationships indicate that the first Samosir lava domes were emplaced to form what today is the Tuk Tuk peninsula and then progressed south along the Samosir fault (Fig. 1). Activity then migrated to the western and southern margin of the caldera. On the western margin at Pusuk Buhit, two lava flows were erupted separately, one at ∼62 ka, followed by a younger lava flow ∼54 ka. In the southern Pardepur region, two post-YTT lava domes are recognized. The first erupted on the caldera rim ∼63 ka on top of older lava flows related to the Oldest Toba Tuff caldera (∼0.8 Ma). The second erupted within the lake ∼57 ka and now forms the Pardepur island. These data indicate the Pardepur lava domes are polygenetic (see Methods section for more details on sample locations). Each spatiotemporal grouping of lava domes is therefore interpreted as a separate pulse of eruptions from the post-climactic magma system.

**Recent resurgent activity.** The eruption of the Samosir lava domes along the Samosir fault, where uplift of at least 700 m is recorded[10,21] (Fig. 1), clearly demonstrates the link between resurgent volcanism and structural uplift of Samosir and suggests that the contribution of magmatism to the uplift of the resurgent dome would have been greatest during the emplacement of the Samosir lava domes[21]. While previous work[21] presented a model where a single pulse of magmatism at 30 ka could have accounted for the temporal pattern of uplift of Samosir, it now appears that episodic magmatism and uplift beginning immediately after the climactic eruption must be considered.

Moreover, radiocarbon dating of sediments found overlying the Tuk Tuk peninsula domes (Supplementary Fig. 1) placed the youngest lake sediments at 8.22 ± 0.04 ka (ref. 21; Supplementary Fig. 2), while the lava dome-lake sediment contact yields an age of 12.77 ± 0.05 ka (Supplementary Fig. 3; Supplementary Data 2). Since zircon eruption ages suggest the extrusion of the Samosir lava domes continued for 4.4 ± 6.4 kyr after the climactic eruption (Fig. 2k) and these ages are distinct from the ages of sediment cover on the domes (Supplementary Fig. 1), we propose the following sequence of events for the development of the Tuk Tuk peninsula.

Soon after the climactic eruption, several polygenetic lobes of lava domes extruded in close proximity, starting with the North Samosir lava dome (∼70 ka) and the Tuk Tuk dome (∼69 ka), followed by a lobe of the West Tuk Tuk lava dome (∼68 ka), and ending with the Bukit Kerbau lava dome (∼65 ka ago). The West Tuk Tuk and Bukit Kerbau lava domes are significantly younger than the YTT (Fig. 2k), therefore these eruptions define a separate episode after the climactic eruption. The earliest sediments on the Tuk Tuk peninsula are dated to 12.77 ± 0.05 ka. This suggests that the Tuk Tuk peninsula was down faulted, causing it to drop below lake level. Lake sediment was then deposited on the Tuk Tuk peninsula from 12.77 ± 0.05 ka to 8.22 ± 0.04 ka (ref. 21) years ago, producing a thick lake sediment package seen on top of Bukit Kerbau (Supplementary Fig. 2). At ∼8 ka, the Tuk Tuk peninsula was again uplifted above lake level, terminating lake sediment deposition with the youngest lake sediment age of 8.22 ± 0.04 ka (ref. 21).

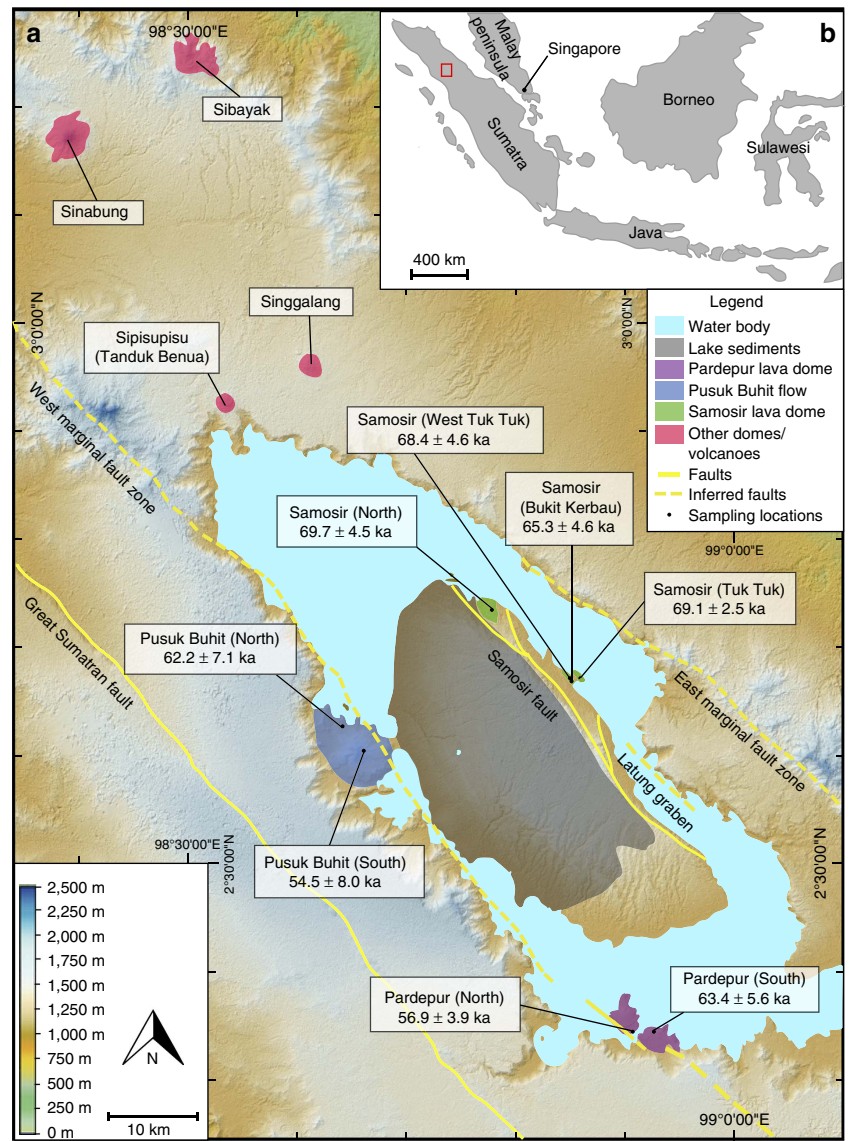

**Figure 1 | Digital elevation map of Toba Caldera and locations of sampled and analysed lava domes.** (**a**) Sample locations and (U–Th)/He eruption ages of analysed lava domes are indicated (green: Samosir domes; blue: Pusuk Buhit flows; purple: Pardepur domes; red: other volcanic extrusions including Sinabung volcano). Inferred fault lines taken from Aldiss and Ghazali[22]. Base map is an Advanced Spaceborne Thermal Emission and Reflection Radiometer (ASTER) 30 m Digital Elevation Model (DEM). (**b**) Overview of Southeast Asia; red rectangle outlines study area of Toba Caldera on the island of Sumatra, Indonesia, as shown in **a**.

**Table 1 | Summary of samples used in this study and their respective eruption ages as calculated using MCHeCalc.**

| Sample name | Sample ID | Rock type | Latitude (N)* | Longitude (E)* | Elevation (m) | Eruption age† (ka) | ±2σ (ka) | MSWD |
|---|---|---|---|---|---|---|---|---|
| YTT (North) | LT_12_001 | Rhyolitic Tuff | 02°59.63 | 098°32.64 | 1,361 | 79.5 | 5.5 | 3.9 |
| YTT (South) | LT_12_012 | Rhyolitic Tuff | 02°39.12 | 098°57.52 | 1,041 | 78.1 | 5.8 | 4.8 |
| Samosir (North) | LT_12_003 | Rhyolitic Lava | 02°44.46 | 098°46.86 | 925 | 69.7 | 4.5 | 3.3 |
| Samosir (Tuk Tuk) | LT_12_008 | Rhyolitic Lava | 02°40.21 | 098°51.42 | 919 | 69.1 | 2.5 | 0.86 |
| Samosir (West Tuk Tuk) | LT_13_014 | Rhyolitic Lava | 02°40.20 | 098°50.72 | 964 | 68.4 | 4.6 | 3 |
| Samosir (Bukit Kerbau) | LT_13_008 | Rhyolitic Lava | 02°39.84 | 098°50.72 | 965 | 65.3 | 4.6 | 2.5 |
| Pusuk Buhit (North) | LT_14_007 | Andesitic Lava | 02°37.76 | 098°38.14 | 1,025 | 62.2 | 7.1 | 3.1 |
| Pusuk Buhit (South) | LT_14_002 | Dacitic Lava | 02°35.91 | 098°38.58 | 1,613 | 54.5 | 8.0 | 5.8 |
| Pardepur (North) | LT_12_016 | Dacitic Lava | 02°20.88 | 098°54.08 | 917 | 56.9 | 3.9 | 3.2 |
| Pardepur (South) | LT_12_015 | Dacitic Lava | 02°20.82 | 098°55.68 | 1,038 | 63.4 | 5.6 | 4.8 |

YTT, Youngest Toba Tuff.
σ represents s.e.
Full data set provided in Supplementary Table 1.
*Lat/Lon in WGS 84 geographical coordinate system.
†Ages based on combined U–Th and (U–Th)/He data (ref. 13).

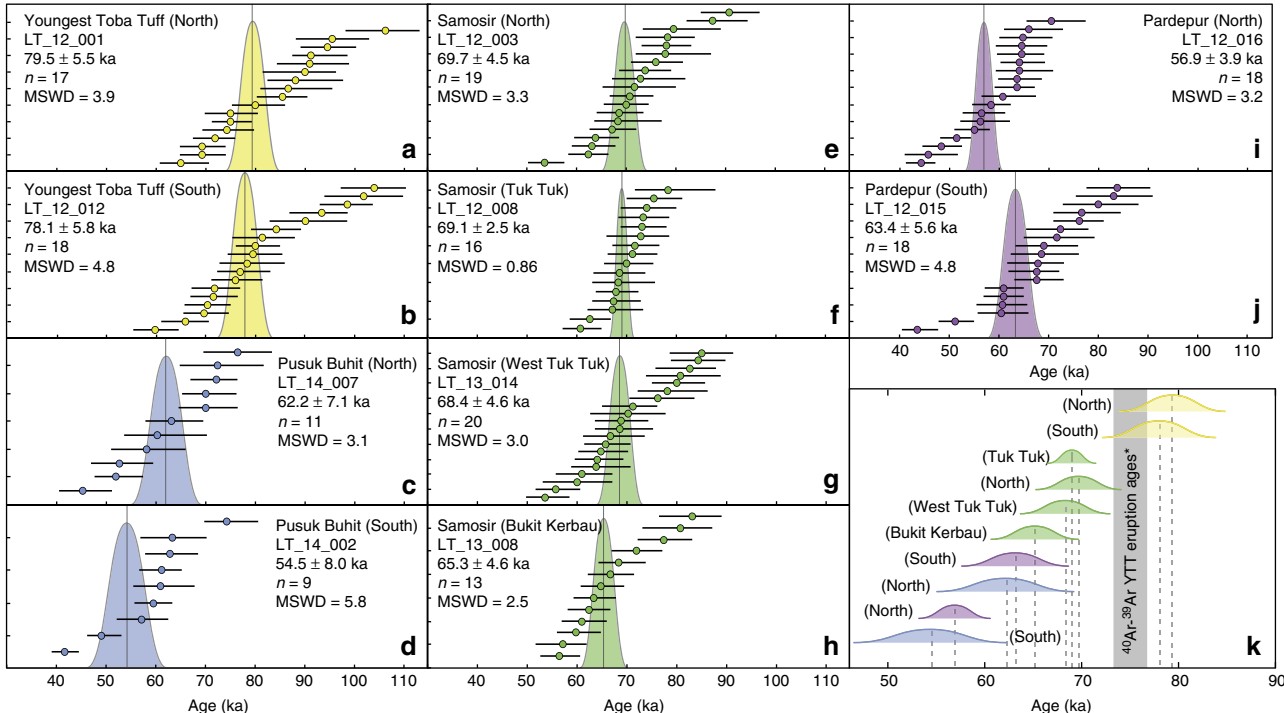

**Figure 2 | Rank-ordered disequilibrium corrected (U–Th)/He zircon ages and evolution of Toba samples.** (**a**–**j**) Rank-ordered disequilibrium corrected (U–Th)/He zircon ages with Gaussian curve of best-fit error-weighted eruption ages for individual YTT and Toba lava dome samples. Individual crystal eruption ages (as points) plotted with ± 2 s.e. (σ) bars. Error-weighted average eruption ages with ± 2σ (as shown by Gaussian curve for each sample) are calculated using MCHeCalc (ref. 13). Plotted data can be found in Supplementary Data 1. (**k**) Evolution of Toba lava domes after YTT eruption, ranked according to (U–Th)/He error-weighted average eruption ages. Grey bar represents range of YTT eruption ages from most recent [40]Ar/[39]Ar analysis of YTT samples[19,20].

Following this event, further subsidence of the eastern portion of the Bukit Kerbau lava dome occurred at 4.27 ± 0.03 ka, based on the age of sediment on top of the lava (Supplementary Fig. 2). An even younger faulting event occurred in the northern section of the Tuk Tuk peninsula, on the West Tuk Tuk lava dome (Supplementary Fig. 3), producing a lava dome-lake sediment contact age of 2.69 ± 0.03 ka. Subsequently, this area was again uplifted above lake level, terminating lake sediment deposition and leaving lake sediments of different ages at a similar elevation height of ~950 m (Supplementary Fig. 3). These constraints indicate that magmatic resurgence at Toba, driving episodic submergence and uplift of the Tuk Tuk peninsula, continued episodically until at least 2.7 ka, much more recently than previously known.

**Sr isotopes and zircon crystallization ages.** The footprint of the Toba magmatic system is extensive, covering the length of lake basins from Pardepur in the south to the northernmost Toba-related centres being Sipisupisu (Tandukbenua) and Singgalang[10]. However, of particular interest here is the active Sinabung volcano, ~30 km to the north of Toba (Fig. 1). Considered to be part of the Sumatran arc and not related to Toba, Sinabung nevertheless shares chemical characteristics with the Toba magmas in contrast to other arc volcanoes. A compilation of new and published data for Sinabung[23–28] show elevated [87]Sr/[86]Sr isotope values from 0.710 to 0.712 overlapping with lithologies related to Toba Caldera (Fig. 3; Supplementary Data 3). Such elevated [87]Sr/[86]Sr compositions > 0.710 are absent from other regional volcanoes from the Sumatran and Sunda–Bandan Arcs. Thus even the least radiogenic Sinabung samples are exceptional on a regional scale. The Sinabung data fall on a

common assimilation–fractional crystallization (AFC) trend with the Toba centres (Fig. 3), further underscoring consanguinity between Sinabung and Toba. This should be a cause for some concern because Sinabung has been in explosive eruption since November 2013, and continues to erupt at the time of writing.

To further test a potential link between Toba and Sinabung, [238]U–[230]Th disequilibrium ages of the outermost rims from the five samples from recent eruptions (including 2014) of Sinabung and ten samples from Toba (YTT and post-caldera domes) were obtained (Fig. 4; Supplementary Data 4). In addition, [238]U–[206]Pb ages of zircon interiors were obtained for zircons found to be in secular equilibrium. Outermost rim ages, measured with the [238]U–[230]Th method, record the last stage of zircon crystallization. For YTT, zircon rim ages range to within few 10,000 years prior to eruption. Previous work shows that some YTT zircon interiors are between 100 and 300 ka (ref. 29). [238]U–[230]Th outermost zircon rim crystallization ages (2σ) for the Sinabung zircons define a population that overlaps YTT and post-caldera dome outermost zircon rim ages dating back to ~350 ka (Supplementary Fig. 4). The Sinabung zircons we analysed show outermost rim ages that overlap with YTT and post-YTT dome zircon rim ages, but extend to younger ages with the youngest Sinabung zircon rim weighted average age at 32.0 ± 14.0 ka (mean square of weighted deviates (MSWD) = 3.6; Fig. 4). The YTT zircons, on the other hand, yield a weighted average rim age of 95.0 ± 6.0 ka (MSWD = 3.6), while the weighted average rim ages for the Pardepur, Samosir and Pusuk Buhit lava domes are 85.6 ± 9.5, 104 ± 5 and 105 ± 19 ka respectively (MSWD = 3.3; 2.7; 1.4 respectively; Fig. 4; Supplementary Data 4).

The data connote that zircon crystallization under Sinabung overlaps in time with the climactic magma system at Toba as well as the post-climactic domes, but has continued well after any

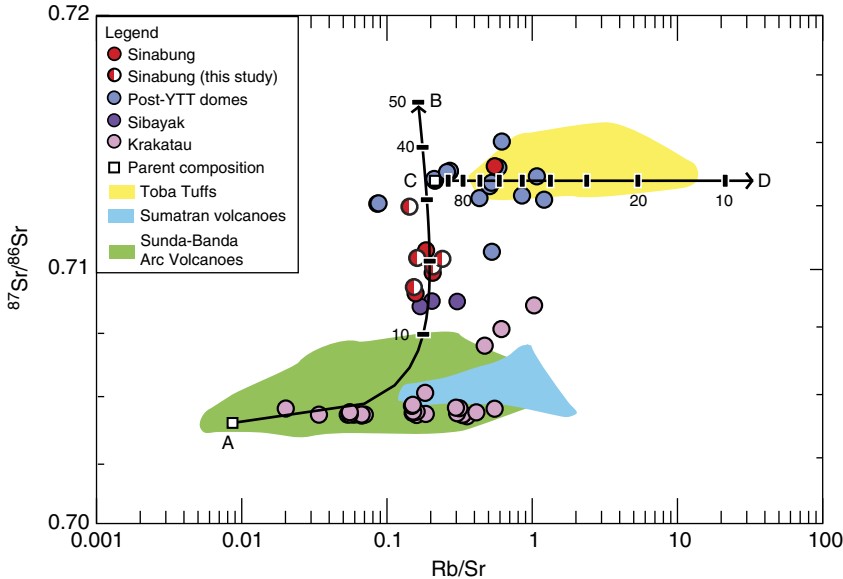

**Figure 3 | Sr-isotope data and AFC model for Sinabung and Toba samples.** AFC model for compiled $^{87}Sr/^{86}Sr$ isotope data from Sinabung and Toba samples[23–28]. Assimilation line is A–B; Parental magma: $^{87}Sr/^{86}Sr = 0.70396$, Rb = 2.6 p.p.m., Sr = 301 p.p.m. Assimilant: $^{87}Sr/^{86}Sr = 0.74036$, Rb = 113 p.p.m., Sr = 32 p.p.m. (chosen based on most enriched $^{87}Sr/^{86}Sr$ granite rock sampled on Sumatra[26]; Supplementary Data 3, Row 560). Fractional Crystallization line is C–D; Parent magma: $^{87}Sr/^{86}Sr = 0.713494$, Rb = 84.1 p.p.m., Sr = 393.4 p.p.m. Daughter magma: $^{87}Sr/^{86}Sr = 0.71383$, Rb = 266 p.p.m., Sr = 24 p.p.m. Parent compositions for A–B and C–D lines graphed as white squares. Plotted data is found in Supplementary Data 3 (half-red circles: Sinabung volcano data from this study; full red circles: previous Sinabung volcano data; blue circles: data for Post-YTT lava domes; purple circles: data for Sibayak volcano; pink circles: data for Krakatau volcano; green field: data for Sunda–Banda Arc volcanoes; blue field: data for Sumatra Arc volcanoes).

known post-caldera eruptions at Toba. These young zircon model ages and the overlapping crystallization history with Toba indicate a shared magmatic system between Sinabung and Toba, rather than a discrete magma reservoir beneath the former. The zircon crystallization history of a small discrete magma body is unlikely to parallel that of a large system like Toba due to the positive feedbacks that enhance the thermal lifetimes, and zircon crystallization histories in the latter[30]. Moreover, the extensive crustal contamination recorded in the Sr-isotope data of Sinabung is also more consistent with a large shared magma system where assimilation would be more efficient and helps explain the anomalous geochemical character of Sinabung compared to the rest of the arc. Our data strongly support the likelihood that the recent eruptions of Sinabung are tapping the Toba magma system, which is active to the present day.

## Discussion

A significant record of structural and magmatic resurgence of Toba Caldera since the climactic eruption of the YTT, ~74 ka, is being revealed through new geochronological data. The new eruption and outermost zircon rim crystallization ages from the post-caldera lava domes and current eruptions at Sinabung reveal that extrusions of lava domes began immediately after the climactic eruption. Combined with the $^{14}C$ ages of sediments on post-caldera lava domes, these data indicate that resurgence at Toba started immediately after the YTT eruption, and eruptions continued episodically for another 6–24 kyr (Fig. 2k). The spatiotemporal pattern of the post-caldera lava domes records spatially discrete pulses of volcanism moving from along the Samosir fault to the southern and western parts of the caldera. The spatial and temporal link between the Samosir lava domes and uplift of Samosir strongly indicates that magmatism associated with the volcanic episodes motivated uplift of the resurgent dome. Furthermore, Toba-like magma has been erupted ~30 km north of Toba from Sinabung volcano where

geochemical affinities with Toba are accompanied by overlapping zircon crystallization histories that continue to more recent times under Sinabung (Fig. 4). The possibility that the Toba magma system is currently being sampled by Sinabung's active and erupting volcano suggests that Toba's magma system extends further north than previous geophysical surveys have indicated[31–33], and should be considered in assessments of hazard associated with activity in this region.

The high-resolution temporal and spatial record of resurgence at Toba provides hitherto unknown details of post-supereruption recovery at supervolcanoes[34–36]. Toba's ~1 million year lifetime is similar to that of other active large resurgent calderas worldwide such as Long Valley, Yellowstone and the Valles Caldera. These systems, like Toba, all have geophysical and petrological evidence of active magmatic systems beneath them[37–40], but are 'older' in that the last supereruptions were at least 630 ka (ref. 41). This has proved challenging for current geochronological techniques to provide insights into the immediate post-supereruption history. Our findings at Toba indicate that magmatism beneath large calderas continues vigorously between supereruptions, motivating structural uplift and feeding hazardous eruptions within and around the caldera almost immediately after the climactic eruption. At older calderas, much of this evidence is probably destroyed or buried by subsequent activity.

Our new geochemical and geochronological data also suggest significant extension of the caldera-related magmatic system well beyond the Toba caldera boundary to beneath Sinabung volcano. While the exact nature of the connection is unknown, a shared magmatic history and magmatic consanguinity is clear and the data are most consistent with a common magmatic reservoir with a shared thermal and chemical evolution. Magmatic complexes that extend well beyond caldera boundaries have been recognized by geophysical surveys at other large calderas[42] and may be more common than previously thought. Like at Yellowstone, at Toba the northern extension may be facilitated by structures that in this case

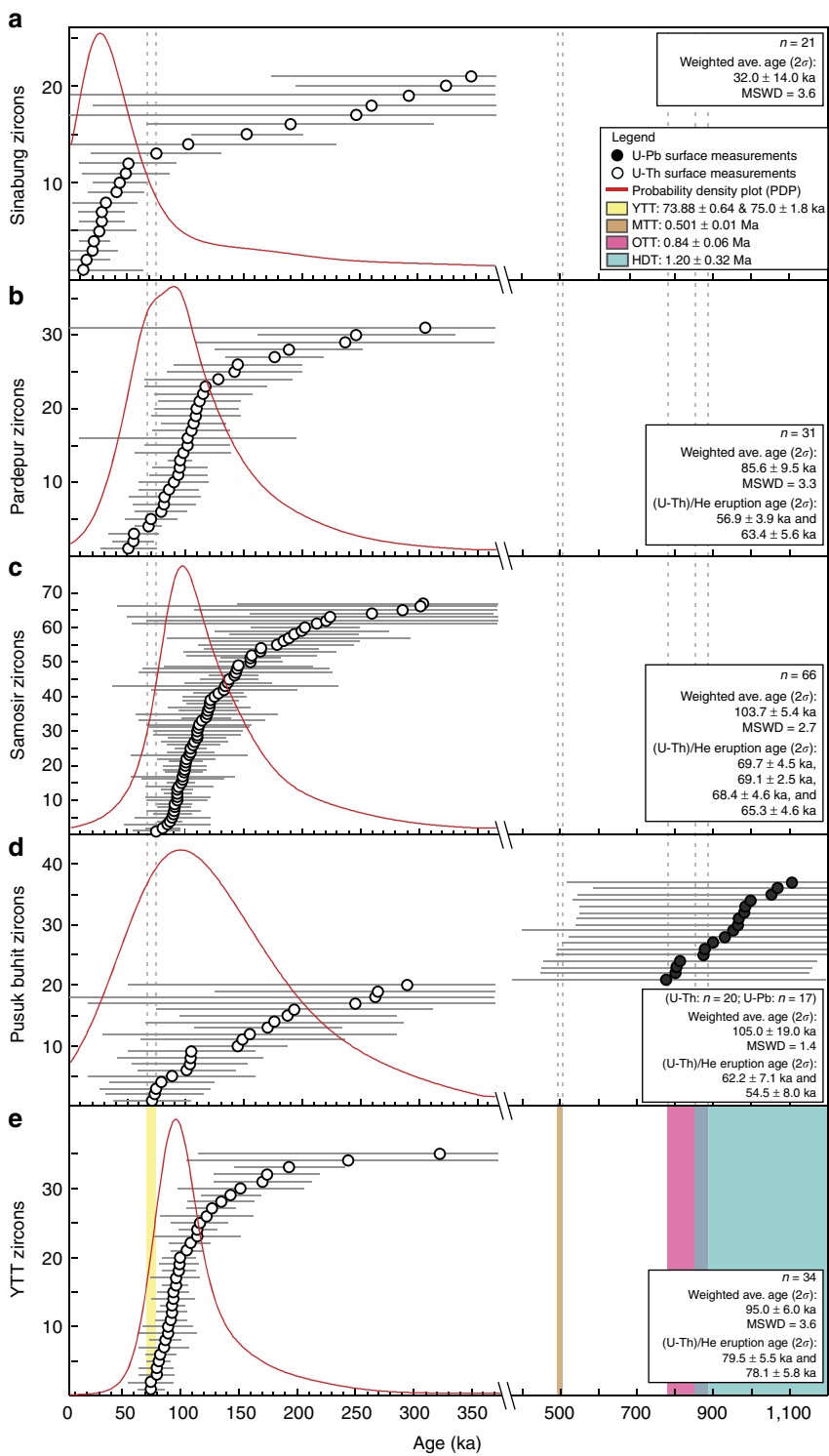

**Figure 4 | $^{238}U$–$^{230}Th$ and $^{238}U$–$^{206}Pb$ ages of surfaces of zircon crystals from Sinabung and Toba.** (**a**–**e**) $^{238}U$–$^{230}Th$ and $^{238}U$–$^{206}Pb$ ages of surfaces of zircon crystals plotted for Sinabung and Toba. (**a**) Shows Sinabung zircons, (**b**–**d**) show post-YTT lava dome zircons from Toba and (**e**) shows YTT zircons. All zircon ages younger than 350 ka (secular equilibrium) are dated with $^{238}U$–$^{230}Th$. Ages older than 350 ka are dated with $^{238}U$–$^{206}Pb$. $^{40}Ar/^{39}Ar$ age range[19,20] for the YTT eruption with ±2 s.e. ($\sigma$) shown by vertical yellow lines. Eruption ages[10] with ±2$\sigma$ of Haranggaol Dacite Tuff (HDT), Oldest Toba Tuff (OTT), Middle Toba Tuff (MTT) shown in aqua, pink, and brown vertical lines, respectively. All errors shown are ±2$\sigma$; *n* refers to the number of grains analysed for each sample; *y*-axis show numbers of zircons for ranked-order plot; see Supplementary Data 4 for complete data set.

parallel the nearby Sumatra fault, along which the Toba depression is thought to have developed as a step-over pull-apart basin[22]. The possibility of Toba's magmatic system extending north to Sinabung raises the need to reassess the potential hazards in this region.

## Methods

**Sampling.** Ten samples from the Toba Caldera complex were collected from four different groups (eight samples from three distinct lava dome/flow regions around Toba Caldera and two YTT samples) in order to determine the eruption age by the combined zircon $^{238}U$–$^{230}Th$ disequilibrium and (U–Th)/He method[13]. Along the

western rim of the caldera, two samples were collected from Pusuk Buhit. Pusuk Buhit is a composite lava dome with noticeable morphological features of lava flows from the summit. The two samples were from two of the stratigraphically youngest lava flows; LT_14_002 (Pusuk Buhit South) from near the summit of Pusuk Buhit, and LT_14_007 (Pusuk Buhit North) from another stratigraphically young lava flow collected at a lower elevation (Table 1).

In the south of the caldera, two samples were collected from the Pardepur lava domes. The Pardepur domes consist of a lava dome extruded along the southern margin of the caldera rim and the small lava dome island, Pulau Sibandang. One sample (LT_12_015; Pardepur South) was collected from near the summit of the caldera rim lava dome, while the second sample (LT_12_016; Pardepur North) was collected from the shoreline of Pulau Sibandang (Table 1).

The third group of lava domes sampled is from Samosir; this includes one sample from the north of Samosir Island, and three samples from the Tuk Tuk peninsula region. The north Samosir Island sample, LT_12_003 (Samosir North), was collected from a road cut around the edge of the lava dome. On the Tuk Tuk peninsula, one sample was collected from the main Tuk Tuk lava dome (LT_12_008; Samosir Tuk Tuk), while the other two (LT_13_014 and LT_13_008; Samosir West Tuk Tuk and Samosir Bukit Kerbau, respectively) were collected from the surrounding peaks (Supplementary Fig. 1).

Five samples from recent activity at Sinabung were collected from exposed outcrop along river cuts on the southern flank of the volcano for zircon $^{238}$U–$^{230}$Th disequilibrium dating[13] and $^{87}$Sr/$^{86}$Sr isotope analysis[43]. The recent two samples were from the 2014 pyroclastic flow, followed by the 2013 lava flow. Two older samples of unknown ages, located lower in the stratigraphic section, were also sampled, one from blocks in a lahar flow and the other from an old lava flow. The fifth sample, material from the 2014 lahar flow, did not yield any zircons but was analysed for $^{87}$Sr/$^{86}$Sr isotopic compositions.

Three sediment samples from lava dome-sediment contact zones on the Tuk Tuk peninsula on Samosir Island (Supplementary Figs 1–3) were analysed by $^{14}$C dating to obtain the age of sediment deposition and provide the minimum age at which the lava dome must have already been extruded.

**Mineral separation.** Samples of pumice and lavas from Toba ignimbrite and lava domes, and volcanic material from Sinabung, were crushed and sieved to the < 250 μm fraction, rinsed, and magnetically separated at currents of 0.25, 0.75, and 1.25 A on a Frantz Isodynamic Separator. The non-magnetic fraction from 1.25 A was run through heavy liquid separation (methylene iodide, 3.33 g cm$^{-3}$) and zircons were hand selected from the denser fraction. Approximately 50 zircon crystals were picked for each Toba sample and then rinsed in cold 48% HF for 1 min to remove adhered glass from zircon surfaces. Sinabung lava flow/pyroclastic material was generally zircon-poor; at least five zircons were picked and processed using the same methodology.

**$^{238}$U–$^{230}$Th analysis.** Subsets of 25 zircons for each sample were selected, mounted, and pressed into a pre-mounted indium (In) mount, exposing the unpolished surfaces of the zircons. Unknown grains were mounted along with secular equilibrium ( > 350 ka) zircon standards AS3 (Duluth Gabbro) for mounts prepared at the University of California—Los Angeles (UCLA) and early-erupted Bishop Tuff for mounts prepared at Stanford University. The surfaces of AS3 were polished to 3 μm before mounting unknown zircons samples. Following the sample mounting, the In mount was coated with a layer of conductive gold and analysed at two different labs: on a CAMECA ims 1,270 secondary ion-mass spectrometer (SIMS or ion microprobe) at UCLA and on the jointly operated Stanford-USGS Sensitive High-Resolution Ion Microprobe-Reverse Geometry (SHRIMP-RG) at Stanford. $^{238}$U–$^{230}$Th zircon analyses followed protocols outlined in Schmitt et al.[13,44,45]

For analyses performed at the UCLA CAMECA ims 1,270, a $^{16}$O$^-$ beam with a primary current of 40–80 nA was focused to a spot size of size of varying from 35 × 40 to 55 × 60 μm on the zircon surface. Measurements were performed using multicollection, and electron-multiplier and Faraday cup detector gains were calibrated by comparing the background-corrected ($^{238}$UO$^+$)/($^{235}$UO$^+$) normalized to ($^{238}$U)/($^{235}$U) = 137.88 (ref. 46). U/Th relative sensitivities were calibrated from analyses of zircon standard 91,500 with 81.2 p.p.m. U and 28.61 p.p.m. Th (ref. 47). The accuracy of the relative sensitivity calibration and background corrections was verified on the AS3 secular equilibrium standard (1,099.1 Ma)[48]. The ($^{230}$Th)/($^{238}$U) weighted averages of AS3 analyses distributed between unknowns was 0.996 ± 0.007 (1 s.d., MSWD = 1.5; n = 17), consistent with secular equilibrium of AS3.

$^{238}$U–$^{230}$Th analyses on SHRIMP-RG at Stanford University were performed with an O$_2^-$ primary beam with intensities ranging from 13 to 15 nA, under similar analytical conditions as on the CAMECA ims 1,270 at UCLA. All isotopes were measured on a single-EPT electron-multiplier operated in pulse counting mode with seven scans (peak-hopping) through the mass table. The ($^{238}$U)/($^{232}$Th) was corrected for instrument mass fractionation using early-erupted Bishop Tuff (767.1 ± 0.9 ka)[49], which is in secular equilibrium. For analyses measured in this session ($^{238}$U)/($^{232}$Th) = 0.911 ± 0.007 (1 s.d., MSWD = 1.04), and all measured values were corrected to ($^{238}$U)/($^{232}$Th) = 1.0. Finally, the zircon $^{238}$U–$^{230}$Th ages were calculated using a two-point isochron through each zircon

analysis and a model melt from the published whole-rock compositions of Toba rocks of ($^{230}$Th/$^{232}$Th) = 0.465 ± 0.002 and ($^{238}$U/$^{232}$Th) = 0.517 ± 0.001 (ref. 50).

**$^{238}$U–$^{206}$Pb analysis.** Zircons found to be in secular equilibrium after initial $^{238}$U–$^{230}$Th analysis were reanalysed for $^{238}$U–$^{206}$Pb on the CAMECA ims 1,270 ion probe SIMS at UCLA; the grains were re-polished and adjacent spots were reanalysed. Analytical conditions (for example, primary current and spot size) were similar to those used for $^{238}$U–$^{230}$Th analyses. $^{238}$U–$^{206}$Pb ages were calculated following the approach of Compston et al.[51] U–Pb relative sensitivities were calibrated using AS3 reference zircon (1,099.1 Ma)[48], which was analysed repeatedly throughout the duration of the analytical session. All reported ages were corrected for common Pb and disequilibrium using D$_{Th}$/U = 0.2. ages > 10 Ma used $^{204}$Pb correction while ages < 10 Ma used $^{207}$Pb correction[46].

Uranium concentrations were measured using (UO$^+$)/(Zr$_2$O$_4^+$), calibrated to 91,500 (81.2 p.p.m. U, 28.61 p.p.m. Th)[47] yielding a relative sensitivity factor of 0.12. Th concentration was calculated based on Th$^+$/U$^+$ relative sensitivity of 0.666, determined on reference zircon AS3 (ref. 52).

**(U–Th)/He analysis.** (U–Th)/He zircon analyses followed protocols outlined in Danišík et al.[53] Zircon crystals were plucked out from the In mounts previously used for SIMS analysis, photographed and measured for dimensions in order to calculate alpha-ejection correction factor[54], and individually transferred into niobium microtubes. Radiogenic $^4$He was extracted at ~ 1,250 °C under ultra-high vacuum using a diode laser and analysed on identical helium extraction lines at the University of Waikato (New Zealand) or at the John de Laeter Centre in Perth (Australia) on a Pfeiffer Prisma QMS-200 mass spectrometer. Released gas was purified using a hot ( ~ 350 °C) Ti–Zr getter, spiked with 99.9% pure $^3$He and introduced into the mass spectrometer next to a cold Ti–Zr getter. $^4$He/$^3$He ratios were measured using a Channeltron detector operated in static mode and corrected for HD and $^3$H interferences by monitoring mass/charge = 1 a.m.u. A 're-extract' was run after each sample to verify complete outgassing of the crystals. He gas results were blank corrected by heating empty Nb tubes using the same procedure. Typically 10–15 blank analyses were measured before and after each crystal measurements.

After the $^4$He measurements, microtubes containing the crystals were retrieved from the laser cell. Following the dissolution procedure of Evans et al.[55], the samples were spiked with $^{235}$U and $^{230}$Th and dissolved in Parr bombs using HF, HNO$_3$ and HCl. Sample, blank and spiked standard solutions were analysed for $^{238}$U, $^{232}$Th and $^{147}$Sm on a Perkin Elmer SCIEX ELAN DRC II ICP-MS (at the University of Waikato) or on an Agilent 7500 ICP-MS at TSW Analytical Ltd (Perth). Total analytical uncertainty of uncorrected (U–Th)/He ages was calculated by propagating uncertainties of U, Th, Sm and He measurements. The uncorrected (U–Th)/He ages were Ft-corrected after Farley et al.[54] assuming a homogeneous distribution of U and Th, verified by the CL images taken on several zircon crystals (Supplementary Fig. 5). Anisotropic diffusion of He in zircon[56] is not an issue in young and quickly cooled volcanic rocks, as in this case for the Toba samples, which essentially have zero residence time in the zircon He partial retention zone[57]. The observed over dispersion of single-grain zircon (U–Th)/He ages indicated by elevated MSWD values (Table 1) is interpreted as resulting from our simplified assumptions regarding zircon crystal geometry and the presence of undetected inclusions that can affect the accuracy of (U–Th)/He ages[58]. The accuracy of zircon (U–Th)/He dating procedure was monitored by replicate analyses of Fish Canyon Tuff zircon (n = 20) measured over the period of this study as internal standard, yielding mean (U–Th)/He age of 28.2 ± 0.8 Ma (1σ), consistent with the reference (U–Th)/He age of 28.3 ± 1.3 Ma (ref. 59).

The Ft-corrected (U–Th)/He ages were then corrected for disequilibrium and pre-eruptive crystal residence by using the MCHeCalc software[44] that requires as input parameters the Ft-corrected zircon (U–Th)/He ages and uncertainties, the zircon crystallization ages and uncertainties, and D$_{230}$ and D$_{231}$ parameters describing zircon-melt fractionation of Th and Pa relative to U. The D$_{230}$ was calculated by dividing measured Th/U ratios of zircons by measured whole-rock Th/U; for D$_{231}$ published Pa/U zircon–rhyolite melt partitioning ratio of 3 was adopted[60]. From resulting distributions of single-grain ages, outliers were rejected based on statistical criteria (interquartile ratio test)[61]. The resulting distribution was tested for normality (with a positive outcome) by using Q–Q-plots and Shapiro–Wilk W-test and used to calculate the final eruption age for each sample as error-weighted average and ± 2 sigma (σ) s.e. uncertainty. For single grain ages that were found to be in secular equilibrium, the age used to represent these grains were the average value of ages for each sample. The maximum 2σ uncertainty from the sample set was used as the 2σ uncertainty for the secular equilibrium single grains. To be conservative with the errors associated with the error-weighted average eruption age, the eruption ages were recalculated in Isoplot 4.15 (ref. 62), using the weighted by assigned/internal errors option. Each eruption age is weighted by the inverse-square of their assigned errors, where the 'probability of fit' is greater than the minimum value defined as 0.15. The 95% confidence is equal to the internal 2σ propagated error multiplied by MSWD and Student's t-test for N-1 degrees of freedom, 95% conf. = τσ√MSWD (Isoplot 4.15) (ref. 62).

**14C analysis.** The organic-rich sediment samples were analysed by accelerator mass spectrometry at Beta Analytic (Supplementary Data 2). Organic sediments were processed in acid wash and results were corrected using the on-line $\delta^{13}C$ accelerator mass spectrometry values of the respective graphite aliquots measured, following instrumental analysis described in Santos et al.[63] Resulting ages are reported both as conventional $^{14}C$ ages (yr BP) as well as 95.4% (2 s.d.) probability calendar age ranges (cal year BP) calibrated by OxCal 4.1 (ref. 64) using the IntCal09 calibration curve[65].

**87Sr/86Sr isotope analysis.** Whole-rock material from the five Sinabung samples were sent to New Mexico State University for $^{87}Sr/^{86}Sr$ isotope analysis by thermal ionization mass spectrometry. Powdered material from each sample was dissolved in a mixture of hydrofluoric, nitric, and hydrochloric acid then purified using chromatographic techniques before analysis on the thermal ionization mass spectrometry, following methodology detailed in Ramos et al.[43]

**Data availability.** The authors declare that all data generated or analysed during this study and supporting the findings of this study are included in this published article and its Supplementary information files. Supplementary Data 1–4 contain all data that was used to generate both the figures in the main text (Figs 1–4) and in the supplementary files (Supplementary Figs 1–5).

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

## Acknowledgements

Fieldwork at Toba and Sinabung and analytical work was supported by NSF EAR 14445634, NSF EAR 1551187, and two Geological Society of America Graduate Student Research Grants. This work was conducted under the auspices of a NSF Graduate Research Fellowship Program under Grant No. 1314109-DGE. M.D. was supported by ARC Discovery funding scheme (DP160102427), the AuScope NCRIS2 program and Australian Scientific Instruments Pty Ltd. M.D. acknowledges the help of C. May, C. Scadding and S. Cameron with solution ICP-MS analyses. The ion microprobe facility at UCLA is partly supported by a grant from the Instrumentation and Facilities Program, Division of Earth Sciences, NSF.

## Author contributions

A.E.M., S.L.d.S. and I.P. conceived the project and conducted the fieldwork; S.L.d.S. and A.E.M. obtained funding. A.E.M., M.D., S.L.d.S., A.K.S. and M.A.C. acquired the age data; M.D. was primarily responsible for (U–Th)/He age data compilation, and A.K.S. and M.A.C. were primarily responsible for $^{238}U–^{230}Th$ disequilibrium data compilation. All authors developed the interpretations presented in the manuscript. A.E.M. took the lead on writing the manuscript, with substantial input by M.D. and S.L.d.S.

## Additional information

**Competing interests:** The authors declare no competing financial interests.

