## [Peer Review File · Nature Communications]

Reviewers' Comments:

Reviewer #1 (Remarks to the Author)

This paper makes several major statements regarding the recent history of the Toba Caldera and its magma chamber. They are based on the acquisition of U-Th/He radiometric ages of post-Youngest Toba Tuff (<74 ka) lava domes and lava flows at Toba, as well as ¹⁴C ages on lake sediments within the caldera, and U-Th crystallization ages of zircon at Sinabung Volcano. These major interpretations include: 1) post YTT eruptive activity began shortly after the climactic eruption at 74 ka and continued for about 15 ka at 3 centers (Samosir Island, Pardepur Island, and Pusuk Buhit Volcano), 2) Caldera resurgence accompanied this period of volcanism but then continued to modern times (2.7 ka), and 3) The Toba magma chamber extends 30 km to the northwest and is presently being tapped by Sinabung Volcano. Although each statement carries its own dramatic appeal, in this reviewer's opinion few of them have been well substantiated by the data presented.

U-Th/He ages

The bulk of the paper relies heavily on the premise that this technique has provided accurate ages of the lavas in question. To provide credibility to the new ages, the authors have dated two samples of YTT and report ages that are 4-5 ka older than the ~74 ka age of the well-dated YTT. Because the 2 sigma error range on these samples encompasses the YTT, the authors conclude that the technique is valid. The lava domes on Samosir, which reportedly are indistinguishable in age from the YTT using the ⁴⁰Ar/³⁹Ar dating technique, produce U-Th/He ages of ~65-70 ka with 2 sigma errors that approach and sometimes overlap with the YTT as well. Using the same argument that the authors have used to prove the validity of the technique (i.e. that the YTT samples were within 2 sigma of the 74 ka age of the YTT), one could argue that because some Samosir lava domes are within 2 sigma of the YTT, that they remain indistinguishable from the YTT.

Some comparative studies between ⁴⁰Ar/³⁹Ar and U-Th/He dating of zircon and olivine (Danišik et al., 2012; Aciego et al., 2010; Blondes et al., 2007) demonstrated systematically younger ages from U-Th/He than for ⁴⁰Ar/³⁹Ar, while another (Schmitt et al., 2011) report overlapping ages. The older ages for the YTT presented in this study are not consistent with several comparative studies and require further explanation. Additionally, anisotropic diffusion of He in zircon has been noted as potential problem with the technique (Reich, et al., 2007). In short, since this is such a major premise upon which many subsequent conclusions depend, I think the authors need to make a more convincing case for the reliability of the U-Th/He technique as applied to intermediate and silicic rocks.

Statement #1 - Post YTT eruptive activity began shortly after the climactic eruption at 74 ka and continued for about 15 ka at 3 centers

Although most of the text relating to this theme of the paper is dependent upon the reliability of the U-Th/He dating technique, some aspects of field work/sampling would be worth noting as well. For example, the authors use the new dates to determine the onset and duration of post-YTT magmatic activity, however, little details are provided regarding the volcanic centers. In the supplemental information file they indicated that Tuk-Tuk was one lava dome that was separated into 4 separate domes by faulting. Was it then a polygenetic dome since the 3 Tuk-Tuk samples vary in age by almost 4 ka? Were the dacite domes at Pardepur monogenetic? Were the two dated samples from the Pusuk Buhit stratovolcano representative of the oldest and youngest flows? The area of active hot springs on Pusuk Buhit might suggest younger eruptive activity than 55 ka, possibly qualifying the 15 ka duration of post-YTT volcanism as a minimum. The authors state that

post-YTT activity began at Tuk-Tuk and "progressed south along the eastern edge of Samosir".....to where? Nothing is indicated on the map. They also suggest that activity progressed westward, yet Tuk-Tuk, Pardepur, and Pusuk Buhit have very close ages whose errors overlap considerably. Instead of separate pulses, one could argue that activity at all 3 centers was concurrent, at least for a while. In short, without additional field data or qualifying statements, the authors are stretching this data set beyond its interpretative limits.

Statement #2 - Caldera resurgence accompanied this period of volcanism but then continued to modern times (2.7 ka)

It seems likely that the Samosir lava domes were associated with the uplift of Samosir Island soon after eruption of the YTT. However, the authors have used the geology of Tuk-Tuk and its lake sediments to suggest that uplift has occurred as recently as 2.7 ka. Because Tuk-Tuk is near lake level, what is the chance that lake level variations are responsible for the sedimentary cover there and not young faulting? What is the field evidence for these faults? Is the 2.7 ka fault scarp well defined? Furthermore, in order to explain the sediment ages (in the supplemental information file), the authors invoke a history of: below-then above-lake level, below-then above-lake level, and below-then above-lake level again. What processes were responsible for such ups and downs? Another problem with this portion of the supplemental text is that the entire first paragraph needs to be re-written to merge Lines 17-21 with the rest of the paragraph. While providing important geological context to the Tuk-Tuk samples, the maps and cross-sections in the Supplementary Information File have some major shortcomings. In Figure 1, the contour lines are unacceptable, extending across the land/lake boundary in several places, and crossing and merging in others. Contour labels are up-side down, and no contour interval is given. In Figure 2, although the line of section crosses "sediment" and "alluvium" none appears in the cross section. I suggest changing "sediment" to "lake sediment" in Figures 2 and 3. Also, in both figures, the vertical exaggeration of the cross sections needs to be stated.

Statement 3 - The Toba magma chamber extends 30 km to the northwest and is presently being tapped by Sinabung Volcano

This study shows that the zircons erupted from recent and older eruptions of Sinabung Volcano have U-Th crystallization ages that overlap with those of the YTT, as well as the Samosir and Pardepur lava domes. The authors state that this, combined with chemical similarities between Sinabung and Toba, indicates that Sinabung is tapping the YTT magma chamber. If so, such a major statement has significant volcanic hazards implications. There are several potential disqualifiers to this bold assertion. First, the authors need to make a convincing case that indeed Sinabung and Toba are chemical counterparts. Three of the four Sinabung sample analyses (from Jones, 1989) used to compare the chemistries of Sinabung and Toba are andesites that are much less radiogenic ($87\text{Sr}/86\text{Sr}$) than any rocks from Toba, including andesites from Sipisupisu on the north rim of Toba Caldera. The other sample from Jones is a rhyolite that appears to be a dead-ringer for the YTT. Since this is the only Sinabung sample that can be shown to overlap geochemically with Toba, the authors should make an attempt to clearly show that it is from Sinabung, and not from the Toba ignimbrite, which extends to Sinabung and may even underlie the modern cone. Chemical affinity arguments aside, Sinabung is still more radiogenic than other Sumatran arc volcanoes. Thus, the magma genesis and evolution processes in the lower crust beneath Sinabung may be similar to those that have occurred beneath Toba, but that does not mean that they have been sourced from the same magma chamber. Furthermore, the authors themselves point out that although geophysical surveys have imaged magma chambers beneath Toba, there is no geophysical evidence to support their conclusion (that Sinabung is tapping an extension of Toba's magma chamber, 30 km north of the caldera rim). In this reviewer's opinion without convincing evidence, such a statement borders on hyperbole.

Other Comments

1. The use of age ranges for the onset and duration of eruptions/uplift is confusing. For example in Lines 103 and 104, it seems that eruption/uplift began ~ 5 ka after the YTT, but a range of 15.7-1.1 ka is given? It would read simpler if the ages themselves, and not their errors were used when describing the chronology.
2. Line 99, "young" Toba Tuff should be "youngest".
3. Line 130 refers to 11 Toba samples, yet Table 1 only shows 10 samples.
4. In Figure 1, the legend should be arranged in stratigraphic order.
5. I'm not quite sure the purpose of having both a "Supplementary Information File" and an "Extended Data" section. Couldn't these be combined into one file?
6. Figure 1 of the Extended Data uses an "assimilant" with a $^{87}\text{Sr}/^{86}\text{Sr}$ ratio of 0.74036. Why was this value chosen?
7. Figure 2 of the Extended Data shows Toba zircons. Why not also include an image of a Sinabung zircon as well?

In summary, this paper presents considerable new and interesting data regarding the post-YTT volcanism at the Toba Caldera and Sinabung Volcano. The main body of the paper is well-written and appropriately referenced. Figures and Tables are reasonable with modifications. However, the major statements/conclusions are not convincing and need to be better substantiated.

Reviewer #2 (Remarks to the Author)

This paper presents a detailed high-resolution chronology of lava dome emplacement after a large caldera-forming eruption. This is novel and significant. We have known for some time that resurgence can occur quickly after a big eruption, but we didn't know exactly how quickly, because the resolving power of the geochronology wasn't sufficient. The major finding of this paper is that the Toba lava domes were emplaced only a few thousand years after the big eruption, and were probably associated with large-scale structural resurgence. This is significant, and I agree with the authors' conclusions.

There are three weaknesses to this paper:

1. The authors argue for discrete pulses of magmatism and discrete magma batches in post-caldera time, but no evidence is presented in support of this hypothesis. What is lacking here is geochemistry. What is needed is a plot or two clearly showing the lavas are geochemically and/or isotopically distinct.
2. There are no geochronological data for Sipisupisu nor Singgalong. In a paper such as this, I feel that this is a major gap.
3. Figure 3 is very interesting but poorly presented and poorly discussed in the paper:
 - i. The y axes are all different, making it impossible for the reader to properly compare the data.
 - ii. Despite this, there appear to be quite different slopes for the tight groupings of data on each plot. However, this is not discussed. I think it is potentially significant.
 - iii. The peak ages of all samples except Sinabung are similar, at 83-96 ka. I actually think the peak for Pardepur is incorrect; it looks more like 90-95 ka. So what is the significance of this very tight grouping, which occurs several tens of thousands of years prior to the main Toba eruption?

Other comments

Line 13: "It has been in a state of resurgence since then". Data from this paper and others do not support this statement. It is important to be clear about what you mean by resurgence. This statement implies that the system has been resurging continually since the last big eruption at ~75 ka, but this is misleading, as later on lines 44 and 72 you state that the resurgence is episodic.

Line 51: Your Toba eruption ages are both older by 3-5 ka than the accepted ages. Might your data have a small systematic offset in them? This needs to be discussed, and it could have an impact on your hiatus calculation on line 59 (also see lines 103-104).

John Stix
3 August 2016

Reviewer #3 (Remarks to the Author)

In summary, the manuscript presents a new analysis involving the timing of post-supereruption activity (or resurgence) at Toba Caldera, Sumatra. The first detailed analysis of resurgent Toba was presented in de Silva et al (2015), which is referenced in this new manuscript. The current study builds on this existing knowledge base with the addition of new zircon geochronology to provide further insight into the timing of post- supereruption activity. The results presented here show that there was an ~8.4 kyr hiatus after the Younger Toba Tuff supereruption before eruptive activity recommenced. New ages for the effusive post-caldera eruptions indicate a general westward progression in eruptive centers across the caldera. In addition, geochemical and geochronological results show that the major (and younger) stratovolcano north of Toba is linked with Toba's magma reservoir. Results highlight that magmatism associated with the voluminous magma reservoir actively continues post-supereruption (with ongoing structural uplift until ~2.7 ka), with resultant implications for volcanic hazard potential.

The methodological approach is novel, in that it uses both zircon U-Th/He and U-Pb or U/Th dating (some of which are surface/outer rim ages which date the final phase of crystallization prior to eruption) to quantify eruption geochronology, in addition to C-14 dating on lava-sediment contacts to constrain the age of recent lava dome extrusion. Data appears to be of good quality and is presented well in figures and supplementary data tables. Uncertainties in age data have been treated appropriately. The conclusions reached seem robust, and are supported by the data presented.

One omission of note, however, is the lack of a reference to the geochemical data supporting the link between Sinabung and Toba magmatic systems (lines 21 and 109) - is this statement referring to Extended Data Table 3 and Extended Data Figure 1? This should then be referenced in the text (around line 109). It is also unclear as to whether this statement is based on new data in the present study, or previously published work (which I assume is the case) - if the latter this needs to be referenced and clarified in the text.

I have made some minor edits and comments in the attached annotated manuscript. In general the manuscript is very well written and concise, and most of my comments are editorial in nature. I have a few queries about the presentation of the various zircon ages ('surface ages' etc.) - a minor rewrite of this section with consistent use of terminology would be helpful for the reader. The figures are well-drafted but would be improved with a few minor changes (see comments in attached manuscript). Supplementary Information Figure 1 could be improved with the addition of an inset overview map, and adding the blue lake colour into the legend. In general appropriate credit has been given to previous work, although I have suggested the addition of at least four

references which are lacking in the final paragraph.

Overall, the manuscript is clear, concise and well-written, with appropriate abstract, introduction and conclusion sections. The results provide a new insight into the Toba magmatic system and will be of interest to the wider geological community, particularly those interested in supervolcano systems and associated magmatic timescales, geochronology, and implications for volcanic hazard assessment. I conclude that the manuscript should be accepted for publication once the suggested minor revisions have been made.

Response to Reviewer's Comments (*Our response In Italics*):

Reviewer #1 (Remarks to the Author):

This paper makes several major statements regarding the recent history of the Toba Caldera and its magma chamber. They are based on the acquisition of U-Th/He radiometric ages of post-Youngest Toba Tuff (<74 ka) lava domes and lava flows at Toba, as well as ¹⁴C ages on lake sediments within the caldera, and U-Th crystallization ages of zircon at Sinabung Volcano. These major interpretations include: 1) post YTT eruptive activity began shortly after the climactic eruption at 74 ka and continued for about 15 ka at 3 centers (Samosir Island, Pardepur Island, and Pusuk Buhit Volcano), 2) Caldera resurgence accompanied this period of volcanism but then continued to modern times (2.7 ka), and 3) The Toba magma chamber extends 30 km to the northwest and is presently being tapped by Sinabung Volcano. Although each statement carries its own dramatic appeal, in this reviewer's opinion few of them have been well substantiated by the data presented.

U-Th/He ages

Comment

The bulk of the paper relies heavily on the premise that this technique has provided accurate ages of the lavas in question. To provide credibility to the new ages, the authors have dated two samples of YTT and report ages that are 4-5 ka older than the ~74 ka age of the well-dated YTT. Because the 2 sigma error range on these samples encompasses the YTT, the authors conclude that the technique is valid. The lava domes on Samosir, which reportedly are indistinguishable in age from the YTT using the ⁴⁰Ar/³⁹Ar dating technique, produce U-Th/He ages of ~65-70 ka with 2 sigma errors that approach and sometimes overlap with the YTT as well. Using the same argument that the authors have used to prove the validity of the technique (i.e. that the YTT samples were within 2 sigma of the 74 ka age of the YTT), one could argue that because some Samosir lava domes are within 2 sigma of the YTT, that they remain indistinguishable from the YTT.

Response

We would like to clarify our view: The validity of the dating technique employed in this study (i.e., combined U-Th-disequilibrium and (U-Th)/He dating of zircon) is not argued here based on the agreement of eruption ages obtained on Younger Toba Tuff (YTT). It has been shown in several independent studies that this technique is a robust geochronometer capable of measuring accurate eruption ages for <350 ka young zircons (for review see e.g. Danišik et al. 2016, Quat. Geochron.). Slight deviations of eruption ages that are commonly observed are related to simplified assumptions of parent nuclides distribution in dated zircons, which is acknowledged in the Supplementary Information Methods section (Lines 91-93). These deviations are reflected in the properly propagated uncertainties associated with the final eruption ages. The fact that our eruption ages obtained for YTT overlap within uncertainty with the eruption ages obtained by $^{40}\text{Ar}/^{39}\text{Ar}$ method in our opinion does prove the credibility of our eruption ages. We are essentially using the YTT as a secondary standard.

With regard to the age of the lava domes, we agree that two of the Samosir lava domes (e.g., North Samosir and Opp Tuk Tuk lava domes) overlap with the YTT and therefore should be treated as indistinguishable from the YTT (text was corrected accordingly). However, Tuk Tuk and Bukit Kerbau lava domes do not overlap with and are clearly younger than the YTT (Figure 2) and therefore cannot be treated as “indistinguishable in age” with YTT.

Comment

Some comparative studies between $^{40}\text{Ar}/^{39}\text{Ar}$ and U-Th/He dating of zircon and olivine (Danišik et al., 2012; Aciego et al., 2010; Blondes et al., 2007) demonstrated systematically younger ages from U-Th/He than for $^{40}\text{Ar}/^{39}\text{Ar}$, while another (Schmitt et al., 2011) report overlapping ages. The older ages for the YTT presented in this study are not consistent with several comparative studies and require further explanation.

Response

We respectfully disagree with this statement and believe that Reviewer 1 is mistaken in arguing for systematic relationship between (U-Th)/He and $^{40}\text{Ar}/^{39}\text{Ar}$ dating methods. These methods rely on different decay schemes, are applied to different minerals and there are numerous studies showing that both methods can yield accurate ages for same rocks. The publications the Reviewer 1 is referring to cannot be cited as evidence for any systematic relationship between (U-Th)/He and $^{40}\text{Ar}/^{39}\text{Ar}$ methods: while the study by Danišík et al. (2012) reported (U-Th)/He ages for Rotoiti tephra that are younger than $^{40}\text{Ar}/^{39}\text{Ar}$ ages on obsidian overlying this tephra in a distal location (Wilson et al., 2007), it is important to realize that both ages are from different rocks (tephra and obsidian, respectively) and therefore cannot be interpreted as showing systematic bias. Moreover, we note that a subsequent study by Flude et al. (2015) reported Ar-Ar ages for sanidine from lithic clasts of the Rotoiti tephra that agree well with (U-Th)/He ages of Danišík et al. (2012). This confirms that the Wilson et al. (2007) $^{40}\text{Ar}/^{39}\text{Ar}$ eruption age for Rotoiti based on obsidian age is indeed “too old”, but also shows that both newer $^{40}\text{Ar}/^{39}\text{Ar}$ and (U-Th)/He results are concordant. The study by Aciego et al. (2010) is not an adequate comparison because it dated olivine which, unlike zircon, is a very low-U mineral and extremely difficult to date by (U-Th)/He method due to its low concentrations of radiogenic Helium. In fact, Aciego et al. (2010) is the only study reporting (U-Th)/He dating of olivine, and therefore this study cannot be used as an argument against reliability of zircon (U-Th)/He geochronometer.

Comment

Additionally, anisotropic diffusion of He in zircon has been noted as potential problem with the technique (Reich, et al., 2007).

Response

Diffusion anisotropy of He in zircon is an experimentally confirmed fact. However, this only has an impact on (U-Th)/He apparent ages for old, slowly cooled samples (e.g., from old cratons) where diffusion anisotropy has to be taken into account if partial loss of He is to be modelled as a thermochronometer. However, as shown by several studies

(e.g. Meesters and Dunai 2002, Chem. Geol.), diffusion is clearly not an issue for young and quickly cooled volcanic rocks, which essentially have zero residence time in the thermal partial retention zone, so this criticism is irrelevant for our samples.

Comment

In short, since this is such a major premise upon which many subsequent conclusions depend, I think the authors need to make a more convincing case for the reliability of the U-Th/He technique as applied to intermediate and silicic rocks.

Response

From the comments above it appears to us that Reviewer 1 has reservations regarding the robustness of the novel approach for dating volcanic eruptions by the combined U-Th disequilibrium and (U-Th)/He dating method used in this study. We would like to direct the Reviewer 1 to our recently published review paper (Danišik et al. 2016, Quat. Geochron.) which explains the principles of this methodology and demonstrates the viability in the combined (U-Th)/He and U-Th disequilibrium dating method in providing accurate eruption ages from many young volcanic systems, as documented by several cross-validation studies. As now stated in the manuscript (Supplementary Information Methods section Lines 94-97), the accuracy of (U-Th)/He method was monitored over the course of this study by analyses of Fish Canyon Tuff zircon, and obtained ages are in excellent agreement with its reference age. We also acknowledge the possibility of age variation related to simplified assumptions of parent nuclides distribution in dated zircons (Supplementary Information Methods section Lines 91-93) and propagate them into associated uncertainties of calculated ages. Given these and our previous results on volcanic rocks, we believe our (U-Th)/He ages are robust, accurate and precise to sufficient extent to warrant firm conclusions regarding the longevity of the post-climactic eruptive activity at Toba caldera. We also emphasize that Ar-Ar dating of Toba post-caldera lavas failed to yield accurate results because of crystal recycling where crystals retained significant amounts of unsupported ⁴⁰Ar (this will be presented and discussed in an AGU abstract this Fall), and although (U-Th)/He ages do not have the same precision, they are currently the only method available to

yield accurate constraints for the eruption ages of these lavas. Finally, we would like to point out that Reviewer 3 has quite the opposite opinion about our methodology and data quality than Reviewer 1 as he/she stated the following: “Data appears to be of good quality and is presented well in figures and supplementary data tables. Uncertainties in age data have been treated appropriately. The conclusions reached seem robust, and are supported by the data presented.”

Statement #1 - Post YTT eruptive activity began shortly after the climactic eruption at 74 ka and continued for about 15 ka at 3 centers

Although most of the text relating to this theme of the paper is dependent upon the reliability of the U-Th/He dating technique, some aspects of field work/sampling would be worth noting as well. For example, the authors use the new dates to determine the onset and duration of post-YTT magmatic activity, however, little details are provided regarding the volcanic centers. In the supplemental information file they indicated that Tuk-Tuk was one lava dome that was separated into 4 separate domes by faulting. Was it then a polygenetic dome since the 3 Tuk-Tuk samples vary in age by almost 4 ka? Were the dacite domes at Pardepur monogenetic? Were the two dated samples from the Pusuk Buhit stratovolcano representative of the oldest and youngest flows? The area of active hot springs on Pusuk Buhit might suggest younger eruptive activity than 55 ka, possibly qualifying the 15 ka duration of post-YTT volcanism as a minimum.

Response

We direct Reviewer 1 to the new Supplemental Information file, where in the Supplementary Discussion section we have expounded on this information from Lines 131-155. The Tuk Tuk lava domes are polygenetic domes, where three separate overlapping lava domes were extruded and subsequently faulted during resurgent uplift. The two Pardepur lava domes are polygenetic; two separate lava domes, one found above older lava flows (older than ~74 ka) along the caldera rim, while the other forming the lava dome island in the south of Lake Toba. The two Pusuk Buhit lava flow samples that were collected were representative of the two youngest lava flows that

were obtainable from the peak of the Pusuk Buhit dome. There are no exposed lava flows or tephra deposits that are younger than our samples. While it is possible that the thermal activity Pusuk Buhit might have been accompanied by eruptive activity, there is no evidence in the form of exposed lava or tephra. Therefore this is a speculation that cannot be addressed. We would like to point out that thermal and hydrothermal activity at volcanoes is often non-magmatic and is rarely associated with eruption of magma. We are confident that the lava flows we have sampled are the youngest erupted at Pusuk Buhit. .

Comment

The authors state that post-YTT activity began at Tuk-Tuk and "progressed south along the eastern edge of Samosir".....to where? Nothing is indicated on the map.

Response

We direct Reviewer 1 to the new Supplemental Information file, where in the Supplementary Discussion section we have expounded on this information from Lines 162-166. Based on our ages, post-YTT activity began with the northern Samosir lava dome (~70 ka) and progressed south, along the Samosir fault, to the Tuk Tuk lava domes (from ~69 to 65 ka). The next stage of post-YTT activity (as indicated by the eruption ages of the lava domes) continued in the south and west of Toba caldera, with lava flows from the Pusuk Buhit dome and the Pardepur lava domes. Thus there is a clear trend of ages southward along the eastern edge of Samosir to and then towards the west side of the caldera. Lines 60-65 in the main text have also been fixed to reflect this clarification.

Comment

They also suggest that activity progressed westward, yet Tuk-Tuk, Pardepur, and Pusuk Buhit have very close ages whose errors overlap considerably. Instead of separate pulses, one could argue that activity at all 3 centers was concurrent, at least for a while. In short, without additional field data or qualifying statements, the authors are stretching this data set beyond its interpretative limits.

Response

Reviewer 1 is correct in stating that the errors of the ages of the different lava domes overlap. However, geochemically, the three Tuk Tuk samples differ both in major and trace elements; the Tuk Tuk sample has the lowest SiO₂ (~70%), Bukit Kerbau with middle SiO₂ (~74%) while the Opp Tuk Tuk sample has the highest SiO₂ (~76%). Moreover, they clearly have distinct histories as evidenced by the presence of absence of sediment cover, and the variable age of any sediment cover. As such, we respectfully stand by our argument that the three Tuk Tuk domes erupted separately and are polygenetic. Some of this detail was presented in de Silva et al, (2015) and new data further support a varied history (Varied history is expounded upon below in response to Statement #2).

Statement #2 - Caldera resurgence accompanied this period of volcanism but then continued to modern times (2.7 ka)

It seems likely that the Samosir lava domes were associated with the uplift of Samosir Island soon after eruption of the YTT. However, the authors have used the geology of Tuk-Tuk and its lake sediments to suggest that uplift has occurred as recently as 2.7 ka. Because Tuk-Tuk is near lake level, what is the chance that lake level variations are responsible for the sedimentary cover there and not young faulting? What is the field evidence for these faults? Is the 2.7 ka fault scarp well defined? Furthermore, in order to explain the sediment ages (in the supplemental information file), the authors invoke a history of: below-then above-lake level, below-then above-lake level, and below-then above-lake level again. What processes were responsible for such ups and downs? Another problem with this portion of the supplemental text is that the entire first paragraph needs to be re-written to merge Lines 17-21 with the rest of the paragraph. While providing important geological context to the Tuk-Tuk samples, the maps and cross-sections in the Supplementary Information File have some major shortcomings.

Response

We refer Reviewer 1 to the Supplementary Information file, Supplementary Discussion section Lines 162-185. With regards to the lake level and whether there have been any variations, we direct Reviewer 1 to de Silva et al. (2015) and their work on showing that the lake level at Toba has been constant since ~72.5 ka (within 1500 years of the YTT eruption). We avoided these details in the original submission because level of detail distracts from the main story presented in Nature Comm and is a nuance that does not change the overall age progression we are presenting here. The evidence for the static lake levels include the lack of any stranded deltas or lake high stands around the lake (also explained in Chesner (2012)). As such, the only possible explanation for lake sediments at a similar elevation height of ~950 m having different ages would be the occurrence of faulting in between the two sample sites. Field evidence clearly shows the sediments are uplifted and domed by the extrusion of the domes and by faulting. There is also evidence in the form of pyroclastic aprons that some of the domes erupted underwater and were then uplifted into the current position. The detailed sequence of events for the eruption and faulting of the lava domes can be found, as mentioned earlier, in the new Supplementary Discussion section.

Comment

In Figure 1, the contour lines are unacceptable, extending across the land/lake boundary in several places, and crossing and merging in others. Contour labels are up-side down, and no contour interval is given. In Figure 2, although the line of section crosses "sediment" and "alluvium" none appears in the cross section. I suggest changing "sediment" to "lake sediment" in Figures 2 and 3. Also, in both figures, the vertical exaggeration of the cross sections needs to be stated.

Response

We thank Reviewer 1 for identifying these issues and refer Reviewer 1 to the Supplementary Figures 1-3. The corrections (with respect to the contours, labeling, legends, and vertical exaggeration) have been made and are reflected in the new figures.

Statement #3 - The Toba magma chamber extends 30 km to the northwest and is presently being tapped by Sinabung Volcano

This study shows that the zircons erupted from recent and older eruptions of Sinabung Volcano have U-Th crystallization ages that overlap with those of the YTT, as well as the Samosir and Pardepur lava domes. The authors state that this, combined with chemical similarities between Sinabung and Toba, indicates that Sinabung is tapping the YTT magma chamber. If so, such a major statement has significant volcanic hazards implications. There are several potential disqualifiers to this bold assertion. First, the authors need to make a convincing case that indeed Sinabung and Toba are chemical counterparts. Three of the four Sinabung sample analyses (from Jones, 1989) used to compare the chemistries of Sinabung and Toba are andesites that are much less radiogenic ($^{87}\text{Sr}/^{86}\text{Sr}$) than any rocks from Toba, including andesites from Sipisupisu on the north rim of Toba Caldera. The other sample from Jones is a rhyolite that appears to be a dead-ringer for the YTT. Since this is the only Sinabung sample that can be shown to overlap geochemically with Toba, the authors should make an attempt to clearly show that it is from Sinabung, and not from the Toba ignimbrite, which extends to Sinabung and may even underlie the modern cone. Chemical affinity arguments aside, Sinabung is still more radiogenic than other Sumatran arc volcanoes. Thus, the magma genesis and evolution processes in the lower crust beneath Sinabung may be similar to those that have occurred beneath Toba, but that does not mean that they have been sourced from the same magma chamber. Furthermore, the authors themselves point out that although geophysical surveys have imaged magma chambers beneath Toba, there is no geophysical evidence to support their conclusion (that Sinabung is tapping an extension of Toba's magma chamber, 30 km north of the caldera rim). In this reviewer's opinion without convincing evidence, such a statement borders on hyperbole.

Response

First we would like to point out that there are two lines of reasoning we use for the link to Toba. The Reviewers only comment on one, the isotopic data. Since the original submission and to further test this connection, and its turns out in anticipation of this

comment, we collected further data on new samples that we had collected in 2014 that are from lava 2010-2014 eruptions (and some older ones). These new data extend from 0.710 to 0.712 (Supplementary Figure 4) and confirm the elevated Sr-isotope composition of Sinabung even in andesitic and dacitic compositions erupted in 2010 and 2014. These data overlap the data for post-YTT lavas at Toba. These are distinct when comparing the Sr-isotopes for the other arc volcanoes as there are no arc volcanoes at 0.710 or higher, so even the least radiogenic Sinabung samples are outliers from the arc volcanoes. They are also more radiogenic than Sinabung's neighbor, Sibayak. Thus the Toba signature is clearly seen at Sinabung. The details of the depth of the signature and the connection are being worked on at present, but this connection is also clear in the zircon data (Figure 3) that the reviewer does not comment on and is discussed in Lines 94-105 in the main text.

The U-Th crystallization ages of Sinabung overlap those at Toba and extend to younger ages than seen at Toba. Since we are dating the surface of the grain, these are not ages of crystals that were assimilated from underlying Toba rocks, but represent bona-fide crystallization ages of Sinabung zircons. Combining the two lines of evidence then suggests that Sinabung has been magmatically active during Toba's activity and it erupts Toba-like magmas that are younger than any activity at Toba. These data are "convincing evidence" in our view and suggest that the hypothesis of a connection with Toba is not hyperbole. While avoiding being alarmist, it is important in our view that this be presented in the context of the new understanding of resurgence at Toba, so that these data and interpretation can inform further studies and hazard assessment at Toba. Please also see our response to Comment #3 of Reviewer 2.

Other Comments

1. The use of age ranges for the onset and duration of eruptions/uplift is confusing. For example in Lines 103 and 104, it seems that eruption/uplift began ~5 ka after the YTT, but a range of 15.7-1.1 ka is given? It would read simpler if the ages themselves, and not their errors were used when describing the chronology.

Response

The range given is from the first Samosir lava dome to the last Pusuk Buhit flow. The age range represents the error-weighted duration of volcanism from our data. The smallest age difference is 1.1 ky and the largest difference is 15.5 ka. The ages data are already given in data tables and figures so, we feel it is best to provide the reader with the calculated duration in this example.

2. Line 99, "young" Toba Tuff should be "youngest".

This has been corrected; now found in Line 51 of the main text.

3. Line 130 refers to 11 Toba samples, yet Table 1 only shows 10 samples.

This has been corrected; now corrected in both Lines 49 and 89 of the main text.

4. In Figure 1, the legend should be arranged in stratigraphic order.

This has been corrected.

5. I'm not quite sure the purpose of having both a "Supplementary Information File" and an "Extended Data" section. Couldn't these be combined into one file?

This has been corrected in accordance with the organization of the manuscript according to the Nature Comm checklist for resubmission.

6. Figure 1 of the Extended Data uses an "assimilant" with a $^{87}\text{Sr}/^{86}\text{Sr}$ ratio of 0.74036. Why was this value chosen?

This value was chosen based on the most enriched Sr-isotope granite rock sampled on Sumatra. Such rocks are found throughout Sumatra and through to comprise a

significant part of the basement in this area (Supplementary Table 3, Row 560; Gasparon and Varne, 1995).

7. Figure 2 of the Extended Data shows Toba zircons. Why not also include an image of a Sinabung zircon as well?

This has been corrected; it can be seen in Supplementary Figure 6.

In summary, this paper presents considerable new and interesting data regarding the post-YTT volcanism at the Toba Caldera and Sinabung Volcano. The main body of the paper is well-written and appropriately referenced. Figures and Tables are reasonable with modifications. However, the major statements/conclusions are not convincing and need to be better substantiated.

Reviewer #2 (Remarks to the Author):

This paper presents a detailed high-resolution chronology of lava dome emplacement after a large caldera-forming eruption. This is novel and significant. We have known for some time that resurgence can occur quickly after a big eruption, but we didn't know exactly how quickly, because the resolving power of the geochronology wasn't sufficient. The major finding of this paper is that the Toba lava domes were emplaced only a few thousand years after the big eruption, and were probably associated with large-scale structural resurgence. This is significant, and I agree with the authors' conclusions.

There are three weaknesses to this paper:

1. The authors argue for discrete pulses of magmatism and discrete magma batches in post-caldera time, but no evidence is presented in support of this hypothesis. What is

lacking here is geochemistry. What is needed is a plot or two clearly showing the lavas are geochemically and/or isotopically distinct.

We refer Reviewer 2 to Supplementary Figure 4 and Supplementary Table 3. We have obtained new Sr isotopes for our Sinabung samples and have included them in the data set.

2. There are no geochronological data for Sipisupisu nor Singgalang. In a paper such as this, I feel that this is a major gap.

Unfortunately, our previous samplings of these two peaks have not produced any zircons because the lavas are quite mafic. We intend to do more detailed sampling in the future.

3. Figure 3 is very interesting but poorly presented and poorly discussed in the paper:

i. The y axes are all different, making it impossible for the reader to properly compare the data.

ii. Despite this, there appear to be quite different slopes for the tight groupings of data on each plot. However, this is not discussed. I think it is potentially significant.

iii. The peak ages of all samples except Sinabung are similar, at 83-96 ka. I actually think the peak for Pardepur is incorrect; it looks more like 90-95 ka. So what is the significance of this very tight grouping, which occurs several tens of thousands of years prior to the main Toba eruption?

Response

We direct Reviewer 2 to our new Figure 3. The figure has been redrafted and the corrections taken into account. The tight grouping of the ages indicates that there was a period of intense zircon crystallization several tens of thousand years before the YTT

eruption. The fact that there is evidence of younger U-Th crystallization ages, the same age as the YTT and younger (for the lava domes and for Sinabung) emphasizes our argument that there is a potential link between the Sinabung system and the Toba system. For Sinabung to have overlapping and similar crystallization age zircons as Toba suggests that Sinabung has had a concordant magmatic history that extends to more recent times than the Toba magma system, and it, uniquely compared to other volcanoes in Sumatra, is erupting magma of the same composition as seen in the Toba system. The peak ages for each sample provided are the error-weighted average ages for the given zircon distribution. As part of the redrafting of this figure, we have changed the scale of the x-axis after 350 ka to provide a clearer picture of the distribution of zircon U-Th crystallization ages. The y-axes show the number of zircons analyzed in the ranked-order plot, and for clarity's sake cannot be scaled the same for all the panels (as the number of zircons analyzed differ for each sample).

Other comments

Line 13: “It has been in a state of resurgence since then”. Data from this paper and others do not support this statement. It is important to be clear about what you mean by resurgence. This statement implies that the system has been resurging continually since the last big eruption at ~75 ka, but this is misleading, as later on lines 44 and 72 you state that the resurgence is episodic.

Response

Toba has been a period of resurgence since the ~75 ka eruption, which includes both structural uplift and volcanic activity. Even when the uplift is episodic (de Silva et al., 2015) the entire time range encompassed is still the period of resurgence.

Line 51: Your Toba eruption ages are both older by 3-5 ka than the accepted ages. Might your data have a small systematic offset in them? This needs to be discussed, and it could have an impact on your hiatus calculation on line 59 (also see lines 103-104).

Response

We respectfully disagree with Reviewer 2. Our YTT ages still overlap with uncertainty with the $^{40}\text{Ar}/^{39}\text{Ar}$ YTT ages, so there is not an issue of systematic offset. For more discussion of this issue, we respectfully direct Reviewer 2 to the discussion paragraph in Reviewer 1's comments, where we put forth our response, stating: "Slight deviations of eruption ages that are commonly observed are related to simplified assumptions of parent nuclides distribution in dated zircons which is acknowledged in the Supplementary Information Methods section (Lines 91-93)."

John Stix

3 August 2016

Reviewer #3 (Remarks to the Author):

In summary, the manuscript presents a new analysis involving the timing of post-supereruption activity (or resurgence) at Toba Caldera, Sumatra. The first detailed analysis of resurgent Toba was presented in de Silva et al (2015), which is referenced in this new manuscript. The current study builds on this existing knowledge base with the addition of new zircon geochronology to provide further insight into the timing of post-supereruption activity. The results presented here show that there was an ~8.4 kyr hiatus after the Younger Toba Tuff supereruption before eruptive activity recommenced. New ages for the effusive post-caldera eruptions indicate a general westward progression in eruptive centers across the caldera. In addition, geochemical and geochronological results show that the major (and younger) stratovolcano north of Toba is linked with Toba's magma reservoir. Results highlight that magmatism associated with the voluminous magma reservoir actively continues post-supereruption (with ongoing structural uplift until ~2.7 ka), with resultant implications for volcanic hazard potential.

The methodological approach is novel, in that it uses both zircon U-Th/He and U-Pb or U/Th dating (some of which are surface/outer rim ages which date the final phase of crystallization prior to eruption) to quantify eruption geochronology, in addition to C-14 dating on lava-sediment contacts to constrain the age of recent lava dome extrusion. Data appears to be of good quality and is presented well in figures and supplementary data tables. Uncertainties in age data have been treated appropriately. The conclusions reached seem robust, and are supported by the data presented.

One omission of note, however, is the lack of a reference to the geochemical data supporting the link between Sinabung and Toba magmatic systems (lines 21 and 109) - is this statement referring to Extended Data Table 3 and Extended Data Figure 1? This should then be referenced in the text (around line 109). It is also unclear as to whether this statement is based on new data in the present study, or previously published work (which I assume is the case) - if the latter this needs to be referenced and clarified in the text.

Response

We refer Reviewer 2 to Supplementary Figure 4, and Supplementary Table 3. We have added new Sr isotope data from our Sinabung samples that only serve to further emphasize this link. The text has been edited according to Reviewer 3's suggestions (Line 90). We also respectfully refer Reviewer 3 to responses on this issue to Reviewer 1 and 2.

I have made some minor edits and comments in the attached annotated manuscript. In general the manuscript is very well written and concise, and most of my comments are editorial in nature. I have a few queries about the presentation of the various zircon ages ('surface ages' etc.) - a minor rewrite of this section with consistent use of terminology would be helpful for the reader. The figures are well-drafted but would be improved with a few minor changes (see comments in attached manuscript). Supplementary Information Figure 1 could be improved with the addition of an inset overview map, and adding the blue lake colour into the legend. In general appropriate credit has been given to previous

work, although I have suggested the addition of at least four references, which are lacking in the final paragraph.

Response

We have made the corrections and changes as mentioned by Reviewer 3. Editorial edits by Reviewer 3 have been taken into account and corrected. References that have been suggested have been included into the main text. Edits to figures in main text and Supplementary Information files have also been corrected. We thank Reviewer 3 for their comments on the Supplementary Figures and refer Reviewer 3 to the Supplementary Information file for the updated Supplementary Discussion text and redrafted Supplementary Figures.

Overall, the manuscript is clear, concise and well-written, with appropriate abstract, introduction and conclusion sections. The results provide a new insight into the Toba magmatic system and will be of interest to the wider geological community, particularly those interested in supervolcano systems and associated magmatic timescales, geochronology, and implications for volcanic hazard assessment. I conclude that the manuscript should be accepted for publication once the suggested minor revisions have been made.